# Effectiveness and protection duration of Covid-19 vaccines and previous infection against any SARS-CoV-2 infection in young adults

Lior Rennert [1] ✉, Zichen Ma[1], Christopher S. McMahan[2] & Delphine Dean [3]

Data on effectiveness and protection duration of Covid-19 vaccines and previous infection against general SARS-CoV-2 infection in general populations are limited. Here we evaluate protection from Covid-19 vaccination (primary series) and previous infection in 21,261 university students undergoing repeated surveillance testing between 8/8/2021–12/04/2021, during which B.1.617 (delta) was the dominant SARS-CoV-2 variant. Estimated mRNA-1273, BNT162b2, and AD26.COV2.S effectiveness against any SARS-CoV-2 infection is 75.4% (95% CI: 70.5-79.5), 65.7% (95% CI: 61.1-69.8), and 42.8% (95% CI: 26.1–55.8), respectively. Among previously infected individuals, protection is 72.9% when unvaccinated (95% CI: 66.1–78.4) and increased by 22.1% with full vaccination (95% CI: 15.8–28.7). Statistically significant decline in protection is observed for mRNA-1273 ($P < .001$), BNT162b2 ($P < .001$), but not Ad26.CoV2.S ($P = 0.40$) or previous infection ($P = 0.12$). mRNA vaccine protection dropped 29.7% (95% CI: 17.9–41.6) six months post- vaccination, from 83.2% to 53.5%. We conclude that the 2-dose mRNA vaccine series initially offers strong protection against general SARS-CoV-2 infection caused by the delta variant in young adults, but protection substantially decreases over time. These findings indicate that vaccinated individuals may still contribute to community spread. While previous SARS-CoV-2 infection consistently provides moderately strong protection against repeat infection from delta, vaccination yields a substantial increase in protection.

[1] Department of Public Health Sciences, Clemson University, Clemson, SC, USA. [2] School of Mathematical and Statistical Sciences, Clemson University, Clemson, SC, USA. [3] Department of Bioengineering, Clemson University, Clemson, SC, USA. ✉email: liorr@clemson.edu

The three authorized vaccines in the United States (US), BNT162b2 (Pfizer-BioNTech), mRNA-1273 (Moderna), and Ad26.COV2.S (Johnson and Johnson-Jansen), have shown strong initial efficacy against symptomatic and severe SARS-CoV-2 infection in randomized clinical trials and in follow-up observational studies both pre- and post-time periods in which B.1.617 (delta) was the dominant SARS-CoV-2 variant[1–16]. However, recent studies have demonstrated a substantial decrease in vaccine protection against SARS-CoV-2 infection[14,17–21]. Based on these data, the US Food and Drug Administration (FDA) authorized boosters for all vaccine recipients ≥18 years of age[19,22].

The majority of studies evaluate vaccine protection against symptomatic or severe SARS-CoV-2 infections in clinical populations. Infections among subclinical populations, which mostly consist of asymptomatic and pre-symptomatic infections[23], are hypothesized to be the primary driver of disease transmission[24]. Inadequate protection against such infections lead to increased community transmission rates and pose direct health risks to both vaccinated and unvaccinated individuals. Therefore, protection against any SARS-CoV-2 infection in subclinical populations is critical for mitigating the spread of Covid-19[23–26]. Data on vaccine protection against any SARS-CoV-2 infection in general, otherwise healthy populations, is limited. Studies evaluating protection and waning immunity from previous SARS-CoV-2 infection are also lacking[27]. Furthermore, because previous SARS-CoV-2 infection provides protection against repeat infection[28], and the majority of SARS-CoV-2 infections are undiagnosed[29], estimates of vaccine effectiveness are relative to unvaccinated individuals rather than unprotected individuals. Failure to adequately account for SARS-CoV-2 infection history may confound estimates of vaccine effectiveness[30].

In this study, we evaluate protection from vaccination, previous infection, and waning immunity against any SARS-CoV-2 infection in a young-adult population within a large public university that has been subject to mandatory high-frequency repeated SARS-CoV-2 surveillance testing since the Fall 2020 semester. Because nearly all SARS-CoV-2 infections were diagnosed, this study setting allows for evaluation of protection from vaccination and previous infection against any SARS-CoV-2 infection. Evaluating protection against any SARS-CoV-2 infection in young adults is especially important for public health. Young adults engage in more frequent and higher density social interactions that increase risk of SARS-CoV-2 exposure and community transmission[31,32], yet have the lowest vaccination rates among adults[33].

## Results
The selection processes for all analytic populations are illustrated in Figs. S1 and S2. Descriptive characteristics for the final study sample are presented in Table 1. A total of 280,223 SARS-CoV-2 tests were conducted on 21,261 individuals during the Fall 2021 follow-up period; average number of SARS-CoV-2 tests per individual during follow-up was 13.18 (SD = 4.50). By the end of follow-up, reported proof of full and partial vaccination was 60.1% and 0.5%, respectively; 30.3% of individuals had no record of vaccination or previous SARS-CoV-2 infection. Key differences among unvaccinated individuals relative to (fully) vaccinated individuals include a higher proportion of males (unvaccinated: 56.1%, vaccinated: 43.9%), higher proportion of White non-Hispanic individuals (unvaccinated: 81.6%, vaccinated: 78.4%), higher proportion of non-residential students (unvaccinated: 71.2%, vaccinated: 65.5%), lower proportion with pre-existing condition (unvaccinated: 4.8%, vaccinated: 5.7%), higher proportion using nicotine or other smoking products (unvaccinated: 8.2%, vaccinated: 4.5%), lower number of overall SARS-CoV-2 tests (unvaccinated: 20.58, vaccinated: 26.15), higher proportion

with a SARS-CoV-2 infection prior to the Fall 2021 semester (unvaccinated: 22.9%, vaccinated: 20.6%), and a higher proportion with a SARS-CoV-2 infection during the Fall 2021 follow-up period (unvaccinated: 12.7%, vaccinated: 4.1%).

Among vaccinated individuals, key differences between vaccine manufactures (Table S1) include proportion who are male (Ad26.CoV2.S: 61.9%, mRNA-1273: 44.8%, BNT162b2: 40.9%), White non-Hispanic (Ad26.CoV2.S: 83.6%, mRNA-1273: 78.2%, BNT162b2: 77.8%), non-residential status (mRNA-1273: 75.9%, Ad26.CoV2.S: 69.6%, BNT162b2: 58.4%), pre-existing conditions (mRNA-1273: 6.6%, Ad26.CoV2.S: 5.8%, BNT162b2: 5.2%), use of nicotine or other smoking products (Ad26.CoV2.S: 6.2%, mRNA-1273: 5.5%, BNT162b2: 3.7%), total number of SARS-CoV-2 tests (mRNA-1273: 28.18, BNT162b2: 25.13, Ad26.CoV2.S: 24.31), SARS-CoV-2 infection prior to the Fall 2021 semester (Ad26.CoV2.S: 24.1%, mRNA-1273: 20.4%, BNT162b2: 20.2%), and SARS-CoV-2 infection during the Fall 2021 follow-up period (Ad26.CoV2.S: 6.5%, BNT162b2: 4.5%, mRNA-1273: 2.9%).

**Protection from vaccination and previous SARS-CoV-2 Infection**. The results for estimated protection are presented in Table 2 and Fig. 1. Overall, vaccine protection against any SARS-CoV-2 infection during the 18-week follow-up period was 67.4% (95% CI: 63.7%,70.7%). The mRNA-1273 vaccine had the highest protection (75.4%, 95% CI: 70.5%,79.5%), followed by BNT162b2 (65.7%, 95% CI: 61.1%,69.8%) and Ad26.COV2.S (42.8%, 95% CI: 26.1%,55.8%); statistically significant differences were found between all three vaccine types (mRNA-1273 vs BNT162b2: $P = 0.001$; mRNA-1273 vs Ad26.COV2.S: $P < 0.001$; BNT162b2 vs Ad26.COV2.S: $P < 0.001$). Compared to individuals with no protection (i.e., unvaccinated without a previous SARS-CoV-2 infection), protection against repeat infection for unvaccinated individuals with a previous SARS-CoV-2 infection was 72.9% (95% CI: 66.1%,78.4%). Protection from both vaccination and previous infection was 95.0% (95% CI: 92.1%,96.9%); i.e., vaccination provided a 22.1% increase in protection among previously infected individuals (95% CI: 15.8%,28.7%).

To assess sensitivity to missing SARS-CoV-2 infection history, analyses were restricted to individuals who participated in Clemson University's Covid-19 surveillance testing since the Fall 2020 semester ($N = 13,057$). Descriptive statistics for this analytic sample are presented in Tables S2 and S3; results are presented in Table S4. Increases in protection were observed for individuals vaccinated with mRNA-1273 (protection: 76.4%, 95% CI: 70.3%,81.3%), BNT162b2 (protection: 68.1%, 95% CI: 61.4%,73.6%), and Ad26.COV2.S (protection: 46.6%, 95% CI: 23.2%,62.8%). Results further excluding individuals vaccinated prior to general eligibility (March 31st, 2021) are provided in Table S5. A mild increase in BNT162b2 effectiveness was observed (protection: 70.0%, 95% CI: 63.1%,75.5%).

The analytic samples for protection against any SARS-CoV-2 infection during the follow-up period between December 28th, 2020 and December 4th, 2021 are presented in Tables S6 and S7, with results presented in Table S8. Moderate increases were observed for BNT162b2 (protection: 69.1%, 95% CI: 62.8%,74.3%), Ad26.COV2.S (protection: 46.6%, 95% CI: 24.2%,62.4%), and previous SARS-CoV-2 infection (protection: 80.9%, 95% CI: 77.1%,84.0%). A mild increase in protection was observed for BNT162b2 (protection: 70.9%, 95% CI: 64.5%,76.2%) when excluding individuals vaccinated prior to March 31st, 2021 (Table S9).

**Waning immunity**. There was a significant increase in the risk of SARS-CoV-2 infection with each month since full vaccination (i.e., 2 weeks past final dose) for mRNA-1273 (HR = 1.25, 95% CI: 1.10,1.42), BNT162b2 (HR = 1.17, 95% CI: 1.09,1.26), but not

**Table 1 Descriptive characteristics for study sample.**

| Characteristic | Total N = 21261 | Fully vaccinated N = 12786 | Unvaccinated N = 8361 | P-value |
|---|---|---|---|---|
| Age: mean (SD) | 20.05 (1.55) | 20.03 (1.56) | 20.07 (1.54) | 0.06* |
| Race/ethnicity: N (%) | | | | <0.001** |
| White, non-Hispanic | 16930 (79.6%) | 10023 (78.4%) | 6824 (81.6%) | <0.001** |
| Black, non-Hispanic | 1283 (6.0%) | 824 (6.4%) | 446 (5.3%) | <0.001** |
| Any race, Hispanic | 1393 (6.6%) | 857 (6.7%) | 526 (6.3%) | 0.25** |
| All other, races non-Hispanic | 1655 (7.8%) | 1082 (8.5%) | 565 (6.8%) | <0.001** |
| Gender: N (%) | | | | <0.001** |
| Female | 10840 (51.0%) | 7153 (55.9%) | 3637 (43.5%) | <0.001** |
| Male | 10364 (48.7%) | 5607 (43.9%) | 4693 (56.1%) | <0.001** |
| Not reported | 57 (0.3%) | 26 (0.2%) | 31 (0.4%) | 0.03** |
| Affiliation: N (%) | | | | <0.001** |
| Residential | 6858 (32.3%) | 4415 (34.5%) | 2412 (28.8%) | <0.001** |
| Non-residential | 14403 (67.7%) | 8371 (65.5%) | 5949 (71.2%) | <0.001** |
| Condition impacting immune response: N (%)♦ | 412 (1.9%) | 238 (1.9%) | 171 (2.0%) | 0.37** |
| Any other pre-existing condition: N (%)■ | 1136 (5.3%) | 731 (5.7%) | 398 (4.8%) | 0.01** |
| High blood pressure | 150 (0.7%) | 88 (0.7%) | 59 (0.7%) | 0.42** |
| Heart disease | 29 (0.1%) | 17 (0.1%) | 12 (0.1%) | 0.72** |
| Diabetes | 111 (0.5%) | 72 (0.6%) | 38 (0.5%) | 1.00** |
| Overweight | 547 (2.6%) | 374 (2.9%) | 172 (2.1%) | 0.38** |
| Kidney disease | 18 (0.1%) | 12 (0.1%) | 6 (0.1%) | 0.85** |
| Cough inefficacy | 5 (0.0%) | 4 (0.0%) | 1 (0.0%) | 0.003** |
| Liver disease | 11 (0.1%) | 8 (0.1%) | 3 (0.0%) | 0.95** |
| Medications▲ N (%) | 480 (2.3%) | 306 (2.4%) | 171 (2.0%) | 0.99** |
| Steroids | 89 (0.4%) | 67 (0.5%) | 21 (0.3%) | 0.33** |
| Chemotherapy | 4 (0.0%) | 4 (0.0%) | 0 (0.0%) | <0.001** |
| Immunosuppressants | 107 (0.5%) | 73 (0.6%) | 34 (0.4%) | 0.77** |
| Use of tobacco or nicotine products: N (%) | 1265 (5.9%) | 576 (4.5%) | 682 (8.2%) | 0.66** |
| SARS-CoV-2 tests per person: mean (SD) | 23.94 (11.73) | 26.15 (11.89) | 20.58 (10.66) | 0.60* |
| Fall 2020 Semester | 4.99 (3.27) | 5.42 (3.35) | 4.30 (3.03) | 0.11* |
| Spring 2021 Semester | 10.51 (5.54) | 11.36 (5.53) | 9.13 (5.30) | 0.004* |
| Fall 2021 Semester¶ | 13.18 (4.50) | 14.21 (3.89) | 11.64 (4.91) | 0.27* |
| Previous SARS-CoV-2 infection: N (%)† | 4572 (21.5%) | 2629 (20.6%) | 1915 (22.9%) | 0.12** |
| SARS-CoV-2 infections during follow-up: N (%)# | 1591 (7.5%) | 518 (4.1%) | 1058 (12.7%) | <0.001** |

♦ Self-reported presence of any condition impacting immune response: HIV, Cancer, Lupus, Rheumatoid Arthritis, Solid organ or bone marrow transplant; sample size may not add to N due to non-selection of specific conditions.
■ Self-reported presence of any of the following conditions: high blood pressure, heart disease, diabetes, overweight or obesity, kidney disease or dialysis, previous stroke or other neurological condition affecting my ability to cough, liver disease, or lung disease; sample size may not add to N due to non-selection of specific conditions.
▲ Self-reported medication use of any of the following: steroids, chemotherapy, immunosuppressants; sample size may not add to N due to non-selection of specific medications.
¶ During follow-up period (8/8/2021–12/04/2021).
† Infection occurring prior to follow-up period (8/7/21).
# % is proportion of individuals within each population infected with SARS-CoV-2 during follow-up period.
* Based on independent two-sample t-test (2-sided p-value).
** Based on chi-squared test for independence.

Ad26.COV2.S (HR = 0.94, 95% CI: 0.81,1.09) or previous SARS-CoV-2 infection (HR = 0.94, 95% CI: 0.87,1.02). Estimated protection over time for the mRNA vaccines is displayed in Fig. 2. Between 0 to 6 months post- vaccination, estimated protection decreased from 88.9% to 58.3% for mRNA-1273 and from 80.2% to 49.2% for BNT162b2. Between 0 to 6 months post vaccination, estimated decrease in overall mRNA vaccine protection against any SARS-CoV-2 infection was 29.7% (95% CI: 17.9%,41.6%). At 6 months post vaccination, statistically significant differences were not observed between mRNA-1273 and BNT162b2 (P = 0.27), mRNA-1273 and Ad26.COV2.S (P = 0.59), and BNT162b2 and Ad26.COV2.S (P = 0.89).

Sensitivity analyses accounting for missing SARS-CoV-2 infection history had a negligible impact on results (Table S10). However, a greater decline was observed when removing individuals vaccinated prior to general eligibility (March 31st, 2021). Estimated hazard ratios for months since vaccination were 1.30 for mRNA-1273 (95% CI: 1.07,1.57) and 1.24 for BNT162b2 (95% CI: 1.07,1.43). Greater declines in mRNA vaccine protection were also observed when restricting the sample to the December 28th, 2020 through December 4th, 2021 follow-up periods (Table S10).

**Classification of vaccination status**. We assess sensitivity to classification of vaccination status by assuming that protection starts at day 7 post vaccination for each dose (as opposed to day 14). In this study, results were not sensitive to the assumption of a 7-day or 14-day period. Estimates of overall protection from vaccination and previous infection are presented in Table S11. Among all estimates, (absolute) differences in protection between 7-day and 14-day classification procedures ranged from 0.0% to 1.2%. Absolute differences in change in monthly SARS-CoV-2 infection risk ranged from 0.00 to 0.02 (Table S12).

**Discussion**

An average of 13.18 SARS-CoV-2 tests were conducted per individual during the Fall 2021 follow-up period. This provided a unique setting to evaluate protection against any SARS-CoV-2 infection from vaccination or previous infection among young adults during a period in which delta was the dominant SARS-CoV-2 variant. We report three important findings from our study. First, Covid-19 vaccines offered protection against any SARS-CoV-2 infection, but strength of protection was dependent on vaccine type. The mRNA-1273 vaccine offered the strongest protection against any SARS-CoV-2 infection (75.4%) throughout the study period, followed by BNT162b2 (65.7%) and Ad26.COV2.S (42.8%). Second, while mRNA vaccines offered strong initial protection against any SARS-CoV-2 infection (83.2%), protection decreased by nearly 30% six months post vaccination (to 53.5%). Furthermore, there were no statistically significant differences in the 6-month effectiveness across the three vaccine types. Third, previous SARS-CoV-2 infection offered moderately strong protection over time against SARS-CoV-2 re-infection caused by the delta variant (72.9%). However, a 22.1% increase in protection was observed for previously

**Table 2 Estimated protection from vaccination and previous infection against SARS-CoV-2 infection between August 8th, 2021 and December 4th, 2021.**

| Vaccine Protection* | # of Individuals | # Positive (%) | Protection: % (95% CI) |
|---|---|---|---|
| Unvaccinated | 8361 | 1058 (12.7%) | *Reference* |
| Fully Vaccinated | 12786 | 518 (4.1%) | 67.4% (63.7–70.7)[a] |
| mRNA-1273 | 4562 | 131 (2.9%) | 75.4% (70.5–79.5)[b] |
| BNT162b2 | 7276 | 325 (4.5%) | 65.7% (61.1–69.8)[b] |
| Ad26.COV2.S | 948 | 62 (6.5%) | 42.8% (26.1–55.8)[b] |
| Protection by Vaccination and Previous Infection History[†] | | | |
| No protection | 6446 | 974 (15.1%) | *Reference* |
| Fully vaccinated | | | |
| No previous infection | 10157 | 500 (4.9%) | 66.2% (62.3–69.7)[c] |
| mRNA-1273 | 3632 | 124 (3.4%) | 74.9% (69.7–79.2)[d] |
| BNT162b2 | 5805 | 314 (5.4%) | 64.5% (59.6–68.8)[d] |
| Ad26.COV2.S | 720 | 62 (8.6%) | 37.2% (18.9–51.4)[¶d] |
| Previous infection | 2629 | 18 (0.7%) | 95.0% (92.1–96.9)[c] |
| mRNA-1273 | 930 | 7 (0.8%) | 94.3% (88.3–97.2)[d] |
| BNT162b2 | 1471 | 11 (0.7%) | 94.5% (90.2–97.0)[d] |
| Ad26.COV2.S | 228 | 0 (0%) | 98.5% (75.7–99.9)[¶d] |
| Previous infection only | 1915 | 84 (4.4%) | 72.9% (66.1–78.4)[d] |

*Protection is relative to unvaccinated individuals.
[†]Protection is relative to individuals with no protection (unvaccinated with no previous SARS-CoV-2 infection).
[¶] Estimates and confidence intervals obtained from Cox regression model with Firth's penalized likelihood method[55].
[a] Estimated via Model 1.1 in Supplementary Note 1.
[b] Estimated via Model 1.2 in Supplementary Note 1.
[c] Estimated via Model 1.3 in Supplementary Note 1.
[d] Estimated via Model 1.4 in Supplementary Note 1.

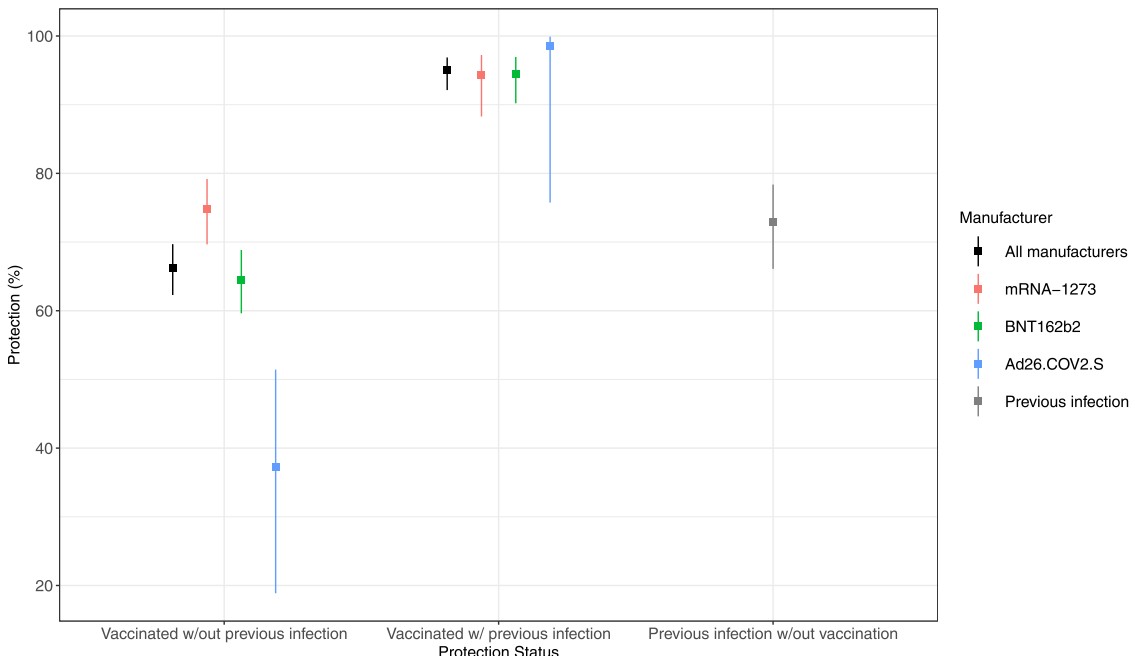

**Fig. 1 Estimated protection from vaccination and previous infection.** Point estimates (with 95% CIs) of (full) vaccine protection among individuals without a previous SARS-CoV-2 infection, vaccine protection among individuals with a previous SARS-CoV-2 infection, and protection from previous infection only. All estimates of protection are relative to individuals with no protection (unvaccinated without a previous SARS-CoV-2 infection).

infected individuals who were fully vaccinated (estimated protection: 95.0%).

Several studies have demonstrated strong protection for the mRNA-1273 and BNT162b2 vaccine against symptomatic or severe SARS-CoV-2 infection caused by the delta variant[6,9–12]. Stronger protection against severe SARS-CoV-2 infection from mRNA vaccines relative to the Ad26.COV2.S vaccine is consistent with other studies[1–4,9,10,12]. The finding that the mRNA-1273

vaccine offered stronger protection relative to the BNT162b2 vaccine is also consistent with several recent studies evaluating protection against severe SARS-CoV-2 infection[9,10,12]. However, direct comparisons between this study and previous studies are not appropriate due to key differences in our study design and setting. First, as opposed to symptomatic or severe SARS-CoV-2 infection, we evaluate protection against any SARS-CoV-2 infection (asymptomatic, pre-symptomatic, and symptomatic).

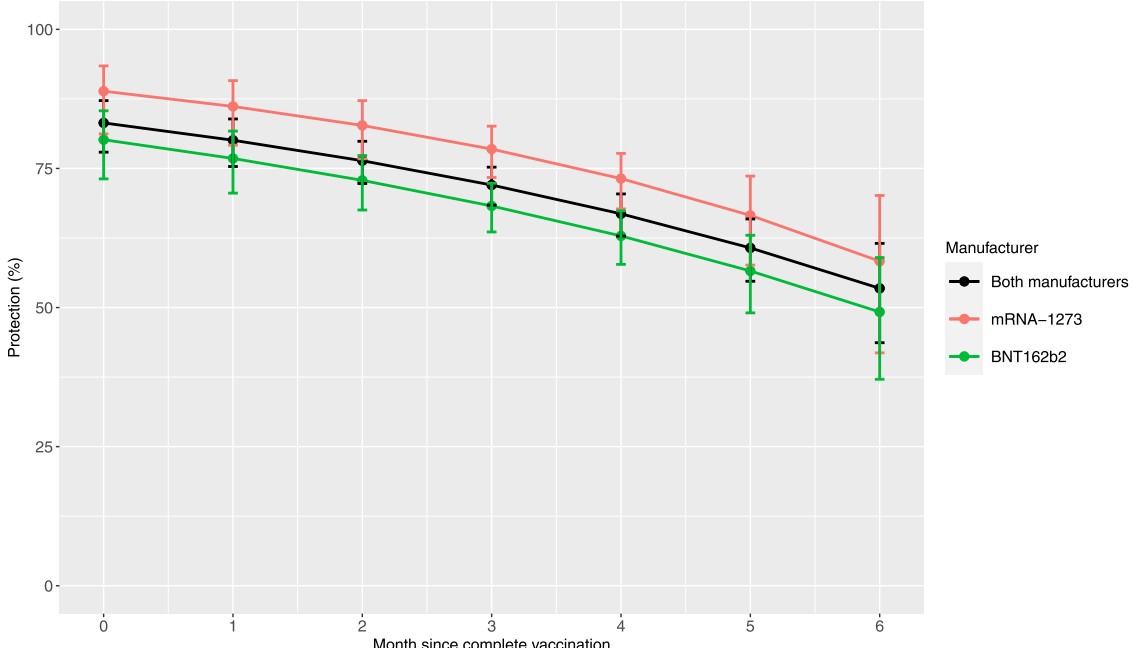

**Fig. 2 Time-varying protection from mRNA vaccination.** Adjusted estimated vaccine protection (point estimate with 95% CIs) by months since full vaccination for mRNA-1273, BNT162b2, and any mRNA vaccine.

Second, our study population consists of young adults in a large university setting. Third, our analysis accounts for SARS-CoV-2 infection history. Omission of this variable may confound estimates of vaccine effectiveness. Furthermore, this allows for estimation of vaccine effectiveness relative to both unvaccinated individuals and individuals with no protection from vaccination or previous SARS-CoV-2 infections.

Our findings have important implications for both individual and community health. While both the mRNA-1273 and BNT162b2 vaccines have shown strong protection against severe SARS-CoV-2 infection, mRNA-1273 showed stronger initial protection against any SARS-CoV-2 infection. However, our results also show a substantial decline in protection over time for both mRNA-1273 and BNT162b2. This was also evidenced in recent studies which found a substantial decrease in BNT162b2 protection against SARS-CoV-2 infection[14,17,18]. While initial protection from Ad26.CoV2.S was substantially lower compared to mRNA protection, we did not observe a decline in Ad26.CoV2.S protection over time. Similar results were demonstrated in a recent study examining antibody responses for these three vaccines[34]. Furthermore, we did not find statistically significant differences in protection six months post vaccination among the three vaccines. While decline in protection against SARS-CoV-2 infection does not necessarily correlate with decreased protection against hospitalization for this age group[14], weakened protection against any SARS-CoV-2 infection may yield increased transmission rates and therefore poses a direct health risk to both vaccinated and unvaccinated individuals. Taken together, these findings provide evidence that booster shots for all three vaccines in this population may both improve individual health and reduce community transmission[20,35–39].

Protection from a previous SARS-CoV-2 infection against reinfection caused by the delta variant was estimated to be 72.9%. Our findings are lower than reported by previous studies[28,40]. However, an important difference in our setting is the longer time period of observation, introduction of delta as the dominant SARS-CoV-2 variant, and assessment of protection against any SARS-CoV-2 infection. This study did not find strong evidence of

waning protection from previous SARS-CoV-2 infection against SARS-CoV-2 re-infection caused by the delta variant. Other studies, prior to introduction of the delta variant, have reached similar conclusions[41,42]. However, recent reports suggest that protection from previous SARS-CoV-2 infection may wane against the Omicron variant[43]. In line with previous studies[7,18,44], our findings also show that for previously infected individuals, vaccination provides a substantial increase in protection against any SARS-CoV-2 infection for all three vaccines receiving emergency use authorization in the United States.

There are several strengths of this study that make it well-suited for evaluating vaccine protection against any SARS-CoV-2 infection. Due to mandated weekly surveillance testing for the majority of the 2021 calendar year in conjunction with temporal dynamics in SARS-CoV-2 viral shedding[45,46], it is likely that a high majority of Covid-19 positive cases, including asymptomatic, pre-symptomatic, and symptomatic cases, were detected. The mandatory testing substantially reduces selection bias that may occur in studies on clinical populations, such as population differences in risk-seeking behaviour[30]. The study sample of relatively homogeneous public university students further reduces the risk of potential confounding. In addition, the main analytic sample was restricted to a period in which delta was the predominant variant, which reduces potential bias that may occur when estimating vaccine effectiveness against multiple variants[11]. Finally, studies evaluating vaccine effectiveness may yield attenuated estimates of protection when previous SARS-CoV-2 infections are not accounted for. The rigorous surveillance testing of this population since the Fall 2020 semester likely captured a majority of individuals previously infected with SARS-CoV-2, which allowed for evaluation of vaccine protection relative to individuals with no protection. In this study sample, over 500,000 SARS-CoV-2 tests were conducted since the Fall 2020 semester, with an average of 24 and 29 tests per individual in the main and secondary analytic samples, respectively.

Our findings are subject to several limitations. This study likely underestimates the true number of previously infected individuals. Surveillance testing was not conducted on a weekly basis

for non-residential students during the Fall 2020 semester and was not mandated between Thanksgiving through the end of 2020. The latter period corresponded to one of the largest Covid-19 surges both statewide and nationally. While symptomatic infections are unlikely subject to severe underreporting since proof of a positive SARS-CoV-2 test provided a 90-day exemption from mandatory testing, asymptomatic infections during this time period were likely underreported. Additional infections may have been missed during the 3-month period of Summer 2021; surveillance testing was not mandated during this window. However, this time frame corresponded to historically low prevalence both statewide and nationally. Additionally, first-year and transfer students did not participate in surveillance testing prior to follow-up and are therefore more likely to have a missing SARS-CoV-2 infection history. However, excluding this sub-population only had a modest impact on estimates of vaccine effectiveness or protection from previous infection.

Misclassification of vaccination status, SARS-CoV-2 infection history, or SARS-CoV-2 exposure during follow-up may bias estimates of protection from vaccination or previous infection[12]. Similarly, underreporting of booster doses would overestimate vaccine effectiveness. Through an external clinic, Clemson University provided free booster doses (for all 3 vaccines) beginning October 29th, 2021. All eligible students had the option to receive the booster while completing their mandatory weekly testing. However, metrics provided by the clinic indicate that only 213 individuals under 25 years of age (includes students, employees, and family members) received a booster dose during the follow-up period, which suggests that the upper bound for misclassification of vaccination status is 1.7%. Furthermore, 94% of student infections during the follow-up period occurred prior to November 2021. Therefore, misclassification of booster doses in this population is expected to have minimal impact on protection estimates.

Another limitation of this study is an inability to differentiate between asymptomatic, pre-symptomatic, or symptomatic infections. Symptom information was only collected from a small sample of individuals prior to testing; no data on symptoms was collected afterwards. This study was not designed to determine biological causes behind the observed differences in mRNA vaccine effectiveness. Relative to BNT162b2, higher observed effectiveness of mRNA-1273 may be attributable to higher mRNA content, longer periods between administration of the first and second dose, or between-group differences that were unaccounted for in this study[8,9].

Finally, time-invariant estimates of vaccine protection must be interpreted with caution. These estimates are dependent on the average time since vaccination in the population. Interpretation of protection in this scenario is specific to both the time at which vaccination occurred in the population and the follow-up period. Therefore, these estimates should not be used to infer protection on the individual level; rather, these should be based on variant-specific estimates of protection that account for time since vaccination.

During an 18-week period in which delta was the dominant SARS-CoV-2 variant, we found that mRNA vaccines offer strong initial protection against any SARS-CoV-2 infection among young adults, but this protection substantially declines with time. The initially high protection from mRNA vaccination against any SARS-CoV-2 infection supports efforts to quickly vaccinate a high majority of the population within a short time period, as this would substantially boost individual health and may quickly mitigate community spread. However, the findings from this study and others suggest that booster doses are needed to restore mRNA vaccine protection against infection to the high levels initially observed after the second dose. The low protection from Ad26.CoV2.S over time suggests that booster doses may be

needed earlier for this population. Previous SARS-CoV-2 infection offered moderately strong protection against any future reinfection from delta and prior variants; however, substantial increases in protection from vaccination were observed among previously infected individuals. Given young adults are major contributors to disease spread yet have some of the lowest vaccination rates, these results support continued efforts to improve Covid-19 vaccination and booster rates in this population.

## Methods

**Study design and population**. This research complies with all relevant ethical regulations. Ethical review for this study was obtained by the Institutional Review Board of Clemson University (IRB # 2021-043-02). Informed consent was waived for this study; students consented to being tested and voluntarily uploaded vaccination information, and de-identified data was for these analyses. In this retrospective cohort study, we compare the rate of SARS-CoV-2 infection between vaccinated individuals, individuals previously infected with SARS-CoV-2, and those with no record of vaccination or previous SARS-CoV-2 infection between August 8th and December 4th, 2021 (follow-up period) at Clemson University in South Carolina (SC). The study sample was restricted to young-adult students between 18 and 24 years of age undergoing surveillance testing during the follow-up period. Student athletes were excluded from this study ($N = 605$), since they represent a higher-risk population and undergo more stringent testing protocols. Also excluded were individuals receiving a vaccine dose without emergency use authorization (EUA) from the U.S. Food and Drug Administration ($N = 102$), individuals with under 21 days between first and second dose of BNT162b2 or under 28 days between first and second dose of mRNA-1273 ($N = 363$), individuals who received two doses from different manufacturers ($N = 2$), and individuals with invalid vaccination cards ($N = 7$). The final study sample consists of 21,261 students undergoing mandatory SARS-CoV-2 surveillance testing during the 18-week follow-up period.

**SARS-CoV-2 testing**. Surveillance testing was primarily conducted through university-provided saliva polymerase-chain-reaction (PCR) tests. Quantification cycle (Cq) values under 33 were considered positive for SARS-CoV-2 (test sensitivity ≥ 95%, test specificity ≥ 99.5%)[47,48]. During the Fall 2021 semester, all students were required to undergo arrival testing between August 8th and August 17th of 2021, followed by weekly surveillance testing beginning at the start of in-person instruction (August 18th, 2021). Testing was mandated for both vaccinated and unvaccinated individuals. Access to campus facilities was restricted for individuals failing to comply with mandatory surveillance testing. SARS-CoV-2 positive individuals were excluded from mandatory surveillance testing for 90 days post infection. Pre-arrival and surveillance testing protocols were similar to the Spring 2021 semester, where all individuals were subject to mandatory weekly testing[28]. Pre-arrival and surveillance testing protocols for both the Fall 2020 and Spring 2021 semesters, including clinical descriptions of testing procedures and additional details on surveillance testing protocols, are described elsewhere[28,47].

**SARS-CoV-2 sequencing**. During August 2021, 180 SARS-CoV-2 positive samples with N1 Cq values less than 30 were randomly selected from a total of 710 saliva isolates collected from Clemson students and employees. These heat-treated positive saliva samples were then sent for sequencing to an external reference lab (Premier Medical Sciences, Greenville SC, USA). RNA was extracted from saliva samples via magnetic beads (Omega) and recovered SARS-CoV-2 RNA quantity was assessed via Codiagnostics Logix smart assay. Samples were processed and sequenced on either an Illumina NovaSeq 6000 or NextSeq500/550 flow cell. Sequences were demultiplexed, assembled, and analyzed in DRAGEN COVID Lineage (Illumina, v.3.5.3). The delta variant was present in 95.6% of sequenced university samples. Between 8/22/2021 and 9/18/2021, it is estimated that the delta variant was present in 98.9% (95% CI: 93.5-99.9%) of SARS-CoV-2 positive samples in SC[49].

**Vaccination status**. Individuals voluntarily uploaded their vaccination status through Clemson University's COVID-19 Voluntary Vaccine Upload Tool. Financial incentives were offered to any individual uploading proof of full vaccination[50]. Furthermore, individuals who were reported as close contacts of SARS-CoV-2 positive cases were exempted from mandatory quarantine with proof of full vaccination. Vaccination data includes vaccine manufacturer and administration dates of first and second vaccine doses. Because protection from vaccination is not immediate[51], vaccination status was lagged by 14 days[12,51,52]. Vaccine status was coded as fully vaccinated if at least 14 days had passed since the second dose of BNT162b2 or mRNA-1273, or 14 days had passed since the first dose of Ad26.COV2.S. Individuals were classified as partially vaccinated if at least 14 days had passed since the first dose of the BNT162b2 or mRNA-1273 vaccine and a second dose was not reported, and unvaccinated otherwise. To assess sensitivity to the assumption of no protection within 14 days post first or second dose, analyses are repeated under the assumption that protection begins at 7 days following each dose[53,54].

**Previous infection status**. Individuals were classified as having previous SARS-CoV-2 infection if they had tested positive as part of university surveillance testing or symptomatic testing, or had uploaded a positive test result through Clemson University's Covid-19 test upload tool. Individuals had incentive to upload positive test results provided by non-University vendors, since this provided a 90-day exemption from mandatory weekly testing.

**Statistical analyses**. Sociodemographic and clinical characteristics by vaccination status are presented in Table 1. Differences between fully vaccinated and unvaccinated individuals were assessed using independent two-sample $t$-tests (2-sided) for continuous variables and chi-squared tests for categorical variables. Differences between vaccine manufacturers were assessed using ANOVA for continuous variables and chi-squared tests for categorical variables. Adjusted time-varying Cox proportional hazard models were used to estimate the relative risk (RR) of SARS-CoV-2 infection by vaccination status during the 18-week follow-up period. Protection (i.e., effectiveness) against any SARS-CoV-2 infection was estimated by 1—RR[28]. The outcome was days between the start of follow-up and date of first SARS-CoV-2 positive test (event date). Individuals who did not test positive during the follow-up period were right-censored at their last negative test date (censoring date). Vaccination status was modelled as a time-varying exposure variable (additional details provided in Supplementary Note 1). Models were adjusted for previous infection, age, race, gender, residential status, comorbidities (measured by three independent binary variables for any immunocompromising pre-existing condition, any other pre-existing condition, and any medication use; additional detail provided in Table 1), and use of nicotine or other smoking products.

To account for the impact of previous infection on vaccine protection, an interaction term between vaccine status and previous infection was included in separate Cox models. In instances of heavy censoring, estimates and confidence intervals obtained were obtained using Firth's penalized likelihood method[55]. In a separate set of analyses, we adjust for time since previous infection when evaluating vaccine protection, and adjust for time since vaccination when evaluate protection from previous infection (Models 1.5-1.8 in Supplementary Note 1). Output from these models is displayed in Table S13. To evaluate waning vaccine protection against SARS-CoV-2 infection, days since vaccination was included in the above Cox models as a time-varying covariate. A similar term was also included to protection over time from previous SARS-CoV-2 infection. Model details are provided in Supplementary Note 2. Finally, while Firth's penalized likelihood method is used only in some models, it was considered for each model.

To assess sensitivity to missing data on SARS-CoV-2 infection history, analyses were repeated for a sample of individuals who participated in Clemson University's Covid-19 surveillance testing since the Fall 2020 semester. This was defined as at least one SARS-CoV-2 surveillance test or SARS-CoV-2 positive test prior to November 25th 2020 (completion of Fall 2020 in-person instruction). Relative to mRNA-1273 and Ad26.COV2.S, a higher proportion of individuals in the analytic sample were vaccinated with BNT162b2 through March 2021, which may bias comparisons between the vaccines. Furthermore, the general adult population was not eligible for vaccination until March 31st, 2021, which introduces potential selection bias through inclusion of higher-risk individuals. We therefore exclude individuals vaccinated prior to March 31st, 2021 in a separate sensitivity analysis.

The analyses above are restricted to a time-period in which delta was the dominant SARS-CoV-2 variant. We also examine vaccine protection against SARS-CoV-2 infection from December 28th, 2020 through December 4th, 2021. The beginning of this follow-up period corresponds to both the roll-out of vaccinations and the start of pre-arrival testing for the Spring 2021 semester. During this period, all individuals were unvaccinated at the study start. All analyses performed in this study were conducted using R version 4.0.5.

**Reporting summary**. Further information on research design is available in the Nature Research Reporting Summary linked to this article.

## Data availability

The raw data are protected and are not available due to data privacy laws. Covid-19 data used in this study can be requested at the following link: https://www.clemson.edu/covid-19/testing/research-data.html. Source data are provided with this paper.

## Code availability

All analyses were run in R version 4.1.2 and utilized the following R packages: tidyverse (version 1.3.1) and survival (version 3.2-13). Code to recreate the data figures in this manuscript, along with sample code to construct the data and run the models, is provided alongside this submission as supplementary material. This code is also available at: https://github.com/ZichenMa-USC/Covid19NatureComm.

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

## Acknowledgements

We thank the Clemson University administration, medical staff, and all other testing providers who helped implement and manage SARS-CoV-2 testing at Clemson University. We thank Clemson's Computing & Information Technology department for their role in collecting, managing, and distributing test results. L.R. and D.D. acknowledge support from US National Institutes of Health (P20 GM121342) during conduct of this study. D.D. acknowledges support from the State of South Carolina (CARES act) for building of the on-campus Covid-19 testing lab and support from US National institutes of Health (R01 MH111366) during conduct of this study. L.R., Z.M., and C.S.M. acknowledge salary support from Clemson University for consulting and modelling work pertaining to development and evaluation of public health strategies (project #1502934).

## Author contributions

L.R. and C.S.M. contributed to the conceptualization of the study. L.R. did the literature search. Z. M. did the figures and visualizations. L.R., Z.M., C.S.M., and D.D. contributed to the methods, L.R., Z.M., and C.S.M. contributed to the results. L.R. and D.D. contributed to the project administration. L.R., Z.M., C.S.M., and D.D. verified the data. L.R. wrote the first draft, and all other authors reviewed the manuscript. The corresponding author had full access to all the data in this study and had final responsibility for the decision to submit for publication.

## Competing interests

The authors declare no competing interests.
