## [Peer Review File · Nature Communications]

Effectiveness and Protection Duration of Covid-19 Vaccines and Previous Infection Against Any SARS-CoV-2 Infection in Young AdultsReviewers' Comments:

Reviewer #1:

Remarks to the Author:

In a student population regularly tested for SARS-CoV-2 infection, protection of vaccination and of previous infections are estimated. Difference between vaccine brands were found, significant decrease in protection for two brands, the added protection of vaccines in previous infected participants was estimated. The regular testing enables taking into account previous infections and broadens the outcome to 'any infection' instead of symptomatic infections as in general settings. This gives valuable addition information compared to other studies.

The methodology and analyses are sound, some improvements can be made.

Major comments

Methods, Statistical Analyses

Looking at the tables, it seems that differences between vaccinated and unvaccinated for categorical variables are tested by dichotomizing for each category and do a separate chi-square test. This is not the appropriate way of testing. Please report the overall p-value of the full chi-square test, do post-hoc tests and report differences for certain categories.

Minor comments

1) Supplied R scripts:

Running `Simulated_Table2` gives error messages when function `EstCI` is called with parameter `"out2"` and `"manufacturer_v23"` or `"manufacturer_v24"`. The values of `manufacturer_v2` in `survival.data` are 0, 1, 2 and 5, so coefficients `manufacturer_v21`, `manufacturer_v22` and `manufacturer_v25` are estimated.

However, changing parameters to `"manufacturer_v21"` or `"manufacturer_v22"` does not give the output as presented in `READ.ME`, but the following

`table2`

```
[,1] [,2] [,3] [,4]
[1,] "Vaccination Protection" "# of Individuals" "# Positive (%)" "Protection: % (95% CI)"
[2,] "Unvaccinated" "1967" "200 (10.2%)" "Reference"
[3,] "Fully Vaccinated" "3001" "269 (9%)" "63.8% (43.4-76.8)"
[4,] "...mRNA-1273" "1058" "94 (8.9%)" "52.9% (39.7-63.2)"
[5,] "...BNT162b2" "1719" "153 (8.9%)" "55.4% (44.7-64.0)"
[6,] "...Ad26.COVS.S" "224" "22 (9.8%)" "63.7% (43.4-76.8)"
[7,] "Protection by Vaccination and Previous Infection History" " " " " " "
[8,] "No protection" "1522" "160 (10.5%)" "Reference"
[9,] "Fully vaccinated" " " " " " "
[10,] "...No previous infection" "2365" "211 (8.9%)" "67.1% (46.0-79.9)"
[11,] ".....mRNA-1273" "841" "73 (8.7%)" "55.2% (40.8-66.1)"
[12,] ".....BNT162b2" "1347" "120 (8.9%)" "59.6% (48.5-68.2)"
[13,] ".....Ad26.COVS.S" "177" "18 (10.2%)" "67.0% (46.0-79.9)"
[14,] "...Previous infection" "636" "58 (9.1%)" "80.2% (46.5-92.7)"
[15,] ".....mRNA-1273" "217" "21 (9.7%)" "79.5% (67.2-87.2)"
[16,] ".....BNT162b2" "372" "33 (8.9%)" "76.2% (65.1-83.8)"
[17,] ".....Ad26.COVS.S" "47" "4 (8.5%)" "80.2% (46.4-92.7)"
[18,] "Previous infection only" "445" "40 (9%)" "63.6% (47.7-74.6)"
>
```

Please check this and if needed also check the script used for the analyses in the paper.

2) Appendix 1 describing the models is overall clear, however, in models 2.1 and 2.2 the presentation and description of the 'time variables', $T_{(V_{(P_i)})}$ etc. raises questions. According to the text in

Methods (top of p.15) and the R script, these are time-varying covariates, so they should be indicated with (t), and their description should be: '..... days between date of and time t'. In the descriptions at model 2.2 the '=1' seems rather strange.

If I am wrong with these comments, please explain.

3) Abstract, Results:

Percentages of (un)vaccinated are 60.1% and 39.3% of the total sample, or 60.5% and 39.5% in the sample excluding partially vaccinated. Please choose one of these options.

4) Abstract, Results:

The sentence "Compared to previously infected individuals who were unvaccinated (protection: 72.9%, 95% CI: 66.1-78.4), a 22.1% increase in protection was observed among previously infected individuals with complete vaccination (95% CI: 15.8-28.7)" is quite hard to read. Consider something like 'Among previously infected individuals the protection was when unvaccinated and increased with '.

5) Introduction

In the sentence "Because nearly all SARS-CoV-2 infections were diagnosed, this study setting allows for evaluation of protection from vaccination and previous infection against any SARS-CoV-2 infection while adequately accounting for SARS-CoV-2 infection history ", previous infection is mentioned both as risk factor and as correction factor, which is confusing.

6) Methods, Statistical Analyses:

Please describe the method used to test differences between the three vaccine brands.

7) Methods, Statistical Analyses, bottom p. 14:

'To account for the impact of previous infection on vaccine protection, an interaction term between vaccine status and previous infection was included in the Cox models.' Please make clear that this is done in a second set of models.

8) Methods, Statistical Analyses, bottom p. 14:

While finally Firth's penalized likelihood method is used only in some models, it will have been considered for each model. Please put the sentence about this at the end of the paragraph.

9) Please describe how the difference in protection at 6 months between vaccine brands is tested, in Methods or in the Appendix.

10) Methods, Statistical Analyses

The sentence 'All analyses were conducted using R version 4.0.5.' is not only related to the last analysis described. Better start a new paragraph, together with the sentence about Firth's penalized likelihood method.

The sentence about the flow charts might go to the Results.

11) Results, p. 5

Rephrase the sentence 'Protection from vaccination among previously infected individuals was 95.0%...' to make clear this protection comes from the combination vaccination plus previous infection.

12) Discussion, p.7 bottom:

The sentence 'Third, our analysis accounts for SARS-CoV-2 infection history, and therefore estimates of vaccine protection (i.e., effectiveness) in this study is relative to populations with no protection from vaccination or previous SARS-CoV-2 infections, rather than unvaccinated populations only.' disregards the two different models, with and without interaction vaccination*infection. The results mentioned in the first paragraph of the Discussion are from model 1.2, this sentence seems to refer to

models 1.3 and 1.4. Please make this clear, e.g. by saying '..... and therefore we could also estimate ... relative to populations with no protection....'

13) Discussion, p.8

The paragraph about international vaccine inequity is certainly interesting, but the relation to the present study is not made clear. Do the authors mean that the waning protection found is or is not reason to supply booster vaccination, or do they only want to say the results may give information for this discussion?

14) Table 1 and other tables:

Are the mean numbers of tests given including participants with no tests in a certain period, or only for the selection with any test?

15) Reconsider the inclusion of the column "Total" in Table S1, S3 and S7.

16) In Table S8 it seems the 'c' for the estimate of Ad26.COVID.S should be a 'b'.

17) Figures S1 and S2:

As the sample in S1 can be expected to be a subgroup of the sample in S2, how can it be that the numbers of some excluded groups are smaller in S2 compared to S1? Were some of these excluded because in the second sample everyone should be unvaccinated at Dec 28, 2020? If so, add this to the flow chart.

General comments:

18) Tables will be more clear when all columns with numbers are right-aligned, and columns with P-values aligned by decimal point.

19) Reconsider the number of decimals in reporting estimate and confidence, see e.g. Cole TJ. Arch Dis Child 2015;100:608-609.

Maria de Ridder

Reviewer #2:

Remarks to the Author:

The manuscript aims to assess the protection afforded by COVID-19 vaccines in young adults. The analysis is based on a mandatory high-frequency repeated SARS-CoV-2 surveillance testing of university students.

The manuscript aims to address three questions:

- 1) Effectiveness of Covid-19 vaccines currently authorized in the United States against any SARS-CoV-2 infection.
- 2) Protection from previous SARS-CoV-2 infection against repeat infection.
- 3) Waning immunity, among 21,261 students (18-24 years) undergoing repeated surveillance testing in a large public university.

General comments:

1. The questions addressed by the investigators were previously addressed by other studies. However, the mandatory high-frequency repeated SARS-CoV-2 surveillance testing of the university students, provides a unique opportunity and a very sound basis for reliable analyses.
2. The authors included three studies in one manuscript. Thus, I will address each one of them separately.
3. The manuscript attempts to address too many questions for one manuscript.

4. It is unclear how the authors estimated that the Delta variant was present in 98.9% of SARS-CoV-2 university students, when the variant was present in 95.6% of the sequenced samples (page 13, lines 311-313).

5. The number of the ethical protocol approved is missing (page 12, line 287).

Study 1: Effectiveness of Covid-19 vaccines currently authorized in the United States against any SARS-CoV-2 infection

Comment 1. The investigators consider as unvaccinated individuals who were really unvaccinated as well as individuals for whom less than 14 days passed from the receipt of the first vaccine dose.

Considering individuals for whom less than 14 days passed as unvaccinated should be regarded as major misclassification. Previous vaccine effectiveness studies demonstrated the following:

A. A protective effect was observed starting day 6 following the first dose (see Chodick G, et al.

Assessment of effectiveness of 1 dose of BNT162b2 vaccine for SARS-CoV-2 infection 13 to 24 days after immunization. *JAMA Netw Open*. 2021 Jun 1;4(6):e2115985)

B. By week three after the first dose vaccine effectiveness against infection was around 50%. Thus it reasonable to assume that some protective effect was present before day 14 (see Glatman-Freedman

A, et al. The BNT162b2 vaccine effectiveness against new COVID-19 cases and complications of breakthrough cases: A nation-wide retrospective longitudinal multiple cohort analysis using individualised data. *EBioMedicine*. 2021 Oct;72:103574.

Comment 2. The investigators performed a cross-sectional evaluation of vaccine effectiveness during a specific evaluation period, without addressing the time passed from the administration of the second vaccine dose. Thus, the only conclusion that can be made from such analysis is that during that period, protection against the circulating variant in the university was suboptimal. Taking into account the time passed from the second vaccine dose, is necessary as part of this portion of the manuscript, irrespective of the fact that waning protection was studied in another portion of the manuscript.

Study 2: Protection from previous SARS-CoV-2 infection against repeat infection.

Comment 1. The investigators performed a cross-sectional evaluation of protection afforded by past infection during a specific evaluation period, without addressing the time passed from the administration of the second vaccine dose.

Comment 2. When combining the effect of infection with vaccination, the time passed from vaccination to new infection was not addressed.

Study 3: Waning immunity, among 21,261 students (18-24 years) undergoing repeated surveillance testing in a large public university.

Comment 1. This study is the strongest study of this manuscript. However, it requires re-classification/re-defining of unvaccinated students (see comment 1 to study 1, above).

REVIEWER COMMENTS

Reviewer #1 (Remarks to the Author):

In a student population regularly tested for SARS-CoV-2 infection, protection of vaccination and of previous infections are estimated. Difference between vaccine brands were found, significant decrease in protection for two brands, the added protection of vaccines in previous infected participants was estimated. The regular testing enables taking into account previous infections and broadens the outcome to 'any infection' instead of symptomatic infections as in general settings. This gives valuable addition information compared to other studies.

The methodology and analyses are sound, some improvements can be made.

Major comments

Methods, Statistical Analyses

Looking at the tables, it seems that differences between vaccinated and unvaccinated for categorical variables are tested by dichotomizing for each category and do a separate chi-square test. This is not the appropriate way of testing. Please report the overall p-value of the full chi-square test, do post-hoc tests and report differences for certain categories.

We thank the Reviewer for the recommendation. An overall p-value from the full chi-squared test has been added. We have made changes to Table 1 in the manuscript, Table S1-S3, S6, S7 in the appendix according to the suggestion.

Minor comments

1) Supplied R scripts:

Running Simulated_Table2 gives error messages when function EstCI is called with parameter "out2" and "manufacturer_v23" or "manufacturer_v24". The values of manufacturer_v2 in survival.data are 0, 1, 2 and 5, so coefficients manufacturer_v21, manufacturer_v22 and manufacturer_v25 are estimated.

However, changing parameters to "manufacturer_v21" or "manufacturer_v22" does not give the output as presented in READ.ME, but the following

table2

```
[,1] [,2] [,3] [,4]
```

```
[1,] "Vaccination Protection" "# of Individuals" "# Positive (%)" "Protection: % (95% CI)"
```

```
[2,] "Unvaccinated" "1967" "200 (10.2%)" "Reference"
```

```
[3,] "Fully Vaccinated" "3001" "269 (9%)" "63.8% (43.4-76.8)"
```

```
[4,] "...mRNA-1273" "1058" "94 (8.9%)" "52.9% (39.7-63.2)"
```

```
[5,] "...BNT162b2" "1719" "153 (8.9%)" "55.4% (44.7-64.0)"
```

```
[6,] "...Ad26.COVS.S" "224" "22 (9.8%)" "63.7% (43.4-76.8)"
```

```
[7,] "Protection by Vaccination and Previous Infection History" " " " " " "
```

```
[8,] "No protection" "1522" "160 (10.5%)" "Reference"
```

```
[9,] "Fully vaccinated" " " " " " "
```

```
[10,] "...No previous infection" "2365" "211 (8.9%)" "67.1% (46.0-79.9)"
```

```

[11,] ".....mRNA-1273" "841" "73 (8.7%)" "55.2% (40.8-66.1)"
[12,] ".....BNT162b2" "1347" "120 (8.9%)" "59.6% (48.5-68.2)"
[13,] ".....Ad26.COVS.S" "177" "18 (10.2%)" "67.0% (46.0-79.9)"
[14,] "...Previous infection" "636" "58 (9.1%)" "80.2% (46.5-92.7)"
[15,] ".....mRNA-1273" "217" "21 (9.7%)" "79.5% (67.2-87.2)"
[16,] ".....BNT162b2" "372" "33 (8.9%)" "76.2% (65.1-83.8)"
[17,] ".....Ad26.COVS.S" "47" "4 (8.5%)" "80.2% (46.4-92.7)"
[18,] "Previous infection only" "445" "40 (9%)" "63.6% (47.7-74.6)"
>

```

Please check this and if needed also check the script used for the analyses in the paper.

We thank the Reviewer for bringing this to our attention. The differences were due to discrepancies in versions of the R packages used. The Demo was run on R version 4.1.2 with R packages tidyverse (version 1.3.1) and survival (version 3.2-13). We have now clarified the R package (and versions) necessary for analysis in the readme file. Furthermore, we have run the code independently on three different machines. All machines have yielded the same output.

2) Appendix 1 describing the models is overall clear, however, in models 2.1 and 2.2 the presentation and description of the ‘time variables’, $T_{(V_{(P_i)})}$ etc. raises questions. According to the text in Methods (top of p.15) and the R script, these are time-varying covariates, so they should be indicated with (t), and their description should be: ‘..... days between date of and time t’. In the descriptions at model 2.2 the ‘=1’ seems rather strange.

If I am wrong with these comments, please explain.

The Reviewer is correct. We have made the appropriate changes to models 2.1 and 2.2 in Appendix 2.

3) Abstract, Results:

Percentages of (un)vaccinated are 60.1% and 39.3% of the total sample, or 60.5% and 39.5% in the sample excluding partially vaccinated. Please choose one of these options.

We thank the Reviewer for catching this discrepancy. We now consistently report these numbers as 60.1% fully vaccinated and 39.3% unvaccinated, in both the Abstract and Results Section.

4) Abstract, Results:

The sentence “Compared to previously infected individuals who were unvaccinated (protection: 72.9%, 95% CI: 66.1-78.4), a 22.1% increase in protection was observed among previously infected individuals with complete vaccination (95% CI: 15.8-28.7)” is quite hard to read. Consider something like ‘Among previously infected individuals the protection was when unvaccinated and increased with’.

We agree with the Reviewer and are grateful for the suggestion. We have taken the Reviewer's advice and have restructured the sentence as follows: "Among previously infected individuals, protection was 72.9% when unvaccinated (95% CI: 66.1-78.4) and increased by 22.1% with complete vaccination (95% CI: 15.8-28.7)."

5) Introduction

In the sentence "Because nearly all SARS-CoV-2 infections were diagnosed, this study setting allows for evaluation of protection from vaccination and previous infection against any SARS-CoV-2 infection while adequately accounting for SARS-CoV-2 infection history ", previous infection is mentioned both as risk factor and as correction factor, which is confusing.

Previous SARS-CoV-2 infection may confound estimates of the association between vaccination and future SARS-CoV-2 infection, since it is associated with both variables. However, we agree with the Reviewer that as currently structured, the above sentence may be confusing to the reader. Upon reflection, we believe that the first part of the sentence adequately accounts for SARS-CoV-2 infection being both a confounder and an estimand of interest. We have therefore removed "while adequately account for SARS-CoV-2 infection history" from this sentence.

6) Methods, Statistical Analyses:

Please describe the method used to test differences between the three vaccine brands.

Differences between vaccine manufacturers were assessed using ANOVA for continuous variables and chi-squared tests for categorical variables. This information is now included in the Statistical Analysis Section.

7) Methods, Statistical Analyses, bottom p. 14:

'To account for the impact of previous infection on vaccine protection, an interaction term between vaccine status and previous infection was included in the Cox models.' Please make clear that this is done in a second set of models.

We have now clarified that these are separate Cox models.

8) Methods, Statistical Analyses, bottom p. 14:

While finally Firth's penalized likelihood method is used only in some models, it will have been considered for each model. Please put the sentence about this at the end of the paragraph.

We thank the Reviewer for the suggestion. We have added this sentence to the end of the 2nd paragraph of the Statistical Analysis Section.

9) Please describe how the difference in protection at 6 months between vaccine brands is tested, in Methods or in the Appendix.

We now describe how differences in protection at 6 months between vaccine manufacturers is tested in Appendix 2, pages 7-8.

10) Methods, Statistical Analyses

The sentence 'All analyses were conducted using R version 4.0.5.' is not only related to the last analysis described. Better start a new paragraph, together with the sentence about Firth's penalized likelihood method. The sentence about the flow charts might go to the Results.

We thank the Reviewer for this suggestion. We have clarified that *all analyses performed in this study were conducted using R version 4.1.12 (previously 4.0.5)*. We have also taken the Reviewer's suggestion and moved the sentence about the selection process and flow charts into the beginning of the Results Section. With regards to the sentence on Firth's penalized likelihood, we have moved it to the appropriate location as suggested by the Reviewer in Comment 8.

11) Results, p. 5

Rephrase the sentence 'Protection from vaccination among previously infected individuals was 95.0%...' to make clear this protection comes from the combination vaccination plus previous infection.

This sentence has been restructured as: "Protection from both vaccination and previous infection was 95.0%..."

12) Discussion, p.7 bottom:

The sentence 'Third, our analysis accounts for SARS-CoV-2 infection history, and therefore estimates of vaccine protection (i.e., effectiveness) in this study is relative to populations with no protection from vaccination or previous SARS-CoV-2 infections, rather than unvaccinated populations only.' disregards the two different models, with and without interaction vaccination*infection. The results mentioned in the first paragraph of the Discussion are from model 1.2, this sentence seems to refer to models 1.3 and 1.4. Please make this clear, e.g. by saying '..... and therefore we could also estimate ... relative to populations with no protection....'

We thank the Reviewer for this suggestion. We have now rewritten this sentence as follows: "Third, our analysis accounts for SARS-CoV-2 infection history. Omission of this variable may confound estimates of vaccine effectiveness. Furthermore, this allows for estimation of vaccine effectiveness relative to both unvaccinated individuals and individuals with no protection from vaccination or previous SARS-CoV-2 infections."

13) Discussion, p.8

The paragraph about international vaccine inequity is certainly interesting, but the relation to the present study is not made clear. Do the authors mean that the waning protection found is or is not reason to supply booster vaccination, or do they only want to say the results may give information for this discussion?

The Reviewer makes a great point. The original intention of this paragraph was to acknowledge that there is a trade-off between supplying booster dose to vaccine-rich countries and primary doses to countries with low

availability. However, upon further reflection, we believe this complicated discussion is beyond the scope of this paper. We have therefore removed this paragraph.

14) Table 1 and other tables:

Are the mean numbers of tests given including participants with no tests in a certain period, or only for the selection with any test?

This is restricted to individuals with at least one test during the follow-up.

15) Reconsider the inclusion of the column "Total" in Table S1, S3 and S7.

We thank the reviewer for catching this mistake. The total column has been relabeled as "Fully vaccinated individuals". The #'s have also been updated. All changes are reflected in the revised appendix.

16) In Table S8 it seems the 'c' for the estimate of Ad26.COV2.S should be a 'b'.

We have made the correction.

17) Figures S1 and S2:

As the sample in S1 can be expected to be a subgroup of the sample in S2, how can it be that the numbers of some excluded groups are smaller in S2 compared to S1? Were some of these excluded because in the second sample everyone should be unvaccinated at Dec 28, 2020? If so, add this to the flow chart.

In the S2 sample in which the follow-up period begins on December 28th, 2020, the discrepancy occurred due to the ordering in which individuals were excluded. Specifically, some of the individuals in the groups with the smaller numbers relative to S1 (student athletes and invalid vaccination cards) were absorbed into the exclusion group for no testing history prior to 11/25/2020. We have re-ordered the filtering criteria to make this consistent with Figure S1 and have updated Figure S2 accordingly (note this has no impact on final sample).

General comments:

18) Tables will be more clear when all columns with numbers are right-aligned, and columns with P-values aligned by decimal point.

We have made the adjustment as requested.

19) Reconsider the number of decimals in reporting estimate and confidence, see e.g. Cole TJ. Arch Dis Child 2015;100:608–609.

The reporting of decimals aligns with the suggested reference.

Reviewer #2 (Remarks to the Author):

The manuscript aims to assess the protection afforded by COVID-19 vaccines in young adults. The analysis is based on a mandatory high-frequency repeated SARS-CoV-2 surveillance testing of university students.

The manuscript aims to address three questions:

- 1) Effectiveness of Covid-19 vaccines currently authorized in the United States against any SARS-CoV-2 infection.
- 2) Protection from previous SARS-CoV-2 infection against repeat infection.
- 3) Waning immunity, among 21,261 students (18-24 years) undergoing repeated surveillance testing in a large public university.

General comments:

1. The questions addressed by the investigators were previously addressed by other studies. However, the mandatory high-frequency repeated SARS-CoV-2 surveillance testing of the university students, provides a unique opportunity and a very sound basis for reliable analyses.
2. The authors included three studies in one manuscript. Thus, I will address each one of them separately.
3. The manuscript attempts to address too many questions for one manuscript.
4. It is unclear how the authors estimated that the Delta variant was present in 98.9% of SARS-CoV-2 university students, when the variant was present in 95.6% of the sequenced samples (page 13, lines 311-313).

The delta variant was present in 95.6% of samples sequenced by the university. In the state of SC, this estimate was 98.9%. We have now modified these sentences for a clearer distinction between the two.

5. The number of the ethical protocol approved is missing (page 12, line 287).

The IRB protocol # is now included.

Study 1: Effectiveness of Covid-19 vaccines currently authorized in the United States against any SARS-CoV-2 infection

Comment 1. The investigators consider as unvaccinated individuals who were really unvaccinated as well as individuals for whom less than 14 days passed from the receipt of the first vaccine dose. Considering individuals for whom less than 14 days passed as unvaccinated should be regarded as major misclassification. Previous vaccine effectiveness studies demonstrated the following:

- A. A protective effect was observed starting day 6 following the first dose (see Chodick G, et al. Assessment of effectiveness of 1 dose of BNT162b2 vaccine for SARS-CoV-2 infection 13 to 24 days after immunization. JAMA Netw Open. 2021 Jun 1;4(6):e2115985)
- B. By week three after the first dose vaccine effectiveness against infection was around 50%. Thus it reasonable to assume that some protective effect was present before day 14 (see Glatman-Freedman A, et al. The BNT162b2 vaccine effectiveness against new COVID-19 cases and complications of breakthrough cases: A nation-wide retrospective longitudinal multiple cohort analysis using individualised data. EBioMedicine. 2021 Oct;72:103574.

We thank the Reviewer for bringing this to our attention, and agree on the importance of properly accounting for the duration period between receipt of vaccination and protection. There appears to be variation in the literature regarding classification of vaccination, with many studies utilizing the 14-day window.¹⁻³ However, regardless of the window, there is bound to be misclassification of vaccination status (whether utilizing the 14-day window and potentially classifying protected individuals as unvaccinated, or utilizing the 7-day window and potentially classifying unprotected individuals as vaccinated).

Given the potential for misclassification and varying definitions in the literature, we have followed the Reviewer's advice and repeated the analyses. Specifically, we assess sensitivity to classification of vaccination status by assuming that protection starts at day 7 post vaccination for each dose (as opposed to day 14). In this study, results were not sensitive to the assumption of a 7-day or 14-day period. Among all estimates of protection, (absolute) differences in protection between classification procedures ranged from 0.0% to 1.2%. For the hazard ratios for change in monthly risk, differences in protection between classification procedures ranged from 0.00 to 0.02. We discuss the sensitivity analyses in the Methods and Results (*Classification of Vaccination Status*), and include the corresponding Tables for this sensitivity analysis in the Appendix: Table S11 and S12 (we are more than happy to oblige if the Reviewer believes this Table should be included in the Main text and/or classified as the primary analysis). We have also updated our Methods Section (Vaccination Status) to include both a 7-day and 14-day window in the classification of vaccination status.

Comment 2. The investigators performed a cross-sectional evaluation of vaccine effectiveness during a specific evaluation period, without addressing the time passed from the administration of the second vaccine dose. Thus, the only conclusion that can be made from such analysis is that during that period, protection against the circulating variant in the university was suboptimal. Taking into account the time passed from the second vaccine dose, is necessary as part of this portion of the manuscript, irrespective of the fact that waning protection was studied in another portion of the manuscript.

The Reviewer raises an important point regarding the interpretation of the results in this setting. Without accounting for time since vaccination, the results must be interpreted as the average vaccine protection in the population at that time (as alluded to by the Reviewer). This is the interpretation given in nearly all studies on Covid-19 vaccine effectiveness. However, it is not possible to account for time since vaccination and give such an interpretation. Rather, when accounting for time since vaccination, the interpretation of vaccine effectiveness is dependent on the time since vaccination. For this reason, we include these in 2 separate models.

To the Reviewer's main point, the interpretation in this setting must be done with caution. We have therefore added the following paragraph to the Discussion Section (at the end of the limitations).

"Finally, time-invariant estimates of vaccine protection must be interpreted with caution. These estimates are dependent on the average time since vaccination in the population. Interpretation of protection in this scenario is specific to both the time at which vaccination occurred in the population and the follow-up period. Therefore, these estimates should not be used to infer protection on the individual level; rather, these should be based on variant-specific estimates of protection that account for time since vaccination."

Study 2: Protection from previous SARS-CoV-2 infection against repeat infection.

Comment 1. The investigators performed a cross-sectional evaluation of protection afforded by past infection during a specific evaluation period, without addressing the time passed from the administration of the second vaccine dose.

The Reviewer is correct that we did not adjust for time since administration of the second dose when estimating the cross-sectional effect of protection from previous infection. Rather, we adjust for this in the models with

time-varying protection. However, because we adjust for time since previous infection in this model, we cannot obtain a “cross sectional” effect of protection from previous infection (in models for time-varying protection).

As requested by the Reviewer, we have therefore repeated the cross-sectional analysis adjusting for time since dose administration. The models for this analysis are detailed in Appendix 1 (Model 1.8) and results are presented in Table S13. Note that this did not have a substantial impact on results: estimated protection from previous infection was 72.9% (95% CI: 66.1-78.4%) in the main analysis and 74.3% (95% CI: 67.4-79.7%) in this sensitivity analysis. In a separate set of models, we adjusted for time since previous infection when estimating protection from vaccination. Across all three vaccines, differences in protection between the main analysis and sensitivity analysis ranged from 0.1% to 0.3%. These findings are also presented in Table S13, with models described in Appendix 1 (Models 1.5 and 1.6). We direct the reader to these analyses in the Statistical Methods Section.

Comment 2. When combining the effect of infection with vaccination, the time passed from vaccination to new infection was not addressed.

Similar to above, we now incorporate time passed from vaccination when estimating the protective effect of previous infection by vaccination status. The models for this analysis are detailed in Appendix 1 (Models 1.7 and 1.8) and results are presented in Table S13. Because estimates of protection in this scenario are specific to time since vaccination, overall estimates of protection are no longer available. Rather, we estimate the hazard ratio (HR) of vaccination at time t + previous infection relative to vaccination at time t + no previous infection. Results were not sensitive to inclusion of time since vaccination. The estimated HR was 0.15 (95% CI: 0.09-0.23) in the main analysis and 0.15 (95% CI: 0.09-0.24) in the sensitivity analyses. When restricting to vaccine manufacturer, differences in HR ranged from 0.00 to 0.01.

Study 3: Waning immunity, among 21,261 students (18-24 years) undergoing repeated surveillance testing in a large public university.

Comment 1. This study is the strongest study of this manuscript. However, it requires re-classification/re-defining of unvaccinated students (see comment 1 to study 1, above).

We thank the Reviewer for the encouragement and agree for the need to carefully define vaccination status. As stated in the response to Comment 1, we have repeated the analyses for waning immunity, this time assuming that protection starts at day 7 (as opposed to day 14). In this setting, changing the window to 7 days did not impact results. Absolute differences in change in monthly SARS-CoV-2 infection risk ranged from 0.00 to 0.02. We have included these additional analyses in Table S12 and discuss the analyses in our Results Section (*Classification of Vaccination Status*).

References

1. Thompson MG, Stenehjem E, Grannis S, et al. Effectiveness of Covid-19 Vaccines in Ambulatory and Inpatient Care Settings. *New England Journal of Medicine*. 2021;0(0):null. doi:10.1056/NEJMoa2110362

2. Tenforde MW. Effectiveness of Pfizer-BioNTech and Moderna Vaccines Against COVID-19 Among Hospitalized Adults Aged ≥ 65 Years — United States, January–March 2021. *MMWR Morb Mortal Wkly Rep.* 2021;70. doi:10.15585/mmwr.mm7018e1
3. Accorsi EK, Britton A, Fleming-Dutra KE, et al. Association Between 3 Doses of mRNA COVID-19 Vaccine and Symptomatic Infection Caused by the SARS-CoV-2 Omicron and Delta Variants. *JAMA*. Published online January 21, 2022. doi:10.1001/jama.2022.0470

Reviewers' Comments:

Reviewer #1:

Remarks to the Author:

I agree with the changes made, except for the presentation of the models in the appendix:
The time-dependent variables T

$i(t)$ and T_V

$i(t)$ and $T_V_C_i(t)$ are correctly presented in models 1.5, 1.6 and 1.7. But in models 1.8, 2.1 and 2.2 the (t) is still lacking. Correct presentation will help the readers understand the analyses done. Line 174 still includes an $'= 1'$.

Reviewer #3:

Remarks to the Author:

This is a review of the manuscript "Effectiveness and Protection Duration of Covid-19 Vaccines and Previous Infection Against Any SARS-CoV-2 Infection in Young Adults" (NCOMMS-21-50698A). Overall, it is a comprehensive paper discussing the vaccine effectiveness (and the duration of this protection) in university students during the delta wave, split by vaccine manufacturer as well as those with/without prior infection and how waning impacts protection against disease.

However, this is a review in place of reviewer #2, to ensure their comments have been covered by the authors in their revised article. In general, all comments by reviewer #2 have been covered or rectified by the authors. Further details are below:

Study 1

RE: comment 1; 14 or 7-day window classification as unvaccinated. This is the biggest concern reviewer 2 had about the article, detailing that a difference between 7 and 14 days may be observed when classifying individuals moving from the unvaccinated > vaccinated group, with a time variable. However, I will agree with the authors that the literature is confounded when classifying individuals as unvaccinated (some use the day of vaccine, whilst others use up to 21 days). Nevertheless, the work the authors have detailed shows that there is a minimal difference when classifying individuals as unvaccinated for 7 or 14 days. The authors have added further clarification to their method and results sections covering this. I do not believe a table needs adding to the main text, the addition of a new section to their results ("Classification of Vaccination Status") summarises this data well and provides the reader with an explanation of the time variable used and how variation by days does not alter the results.

RE: comment 2; time passed since 2nd dose. The addition of a further paragraph within the discussion satisfies reviewer #2 comments, and expands on the limitations (e.g. not individual-level protection).

Study 2

RE: comment 1; adjusting for time since dose. Again, the authors have shown that their analysis has been sufficient to answer this question, detailing minimal differences between their original estimated protection or sensitivity analysis. The authors have included this data in model 1.8 and table S13.

RE: comment 2; time passed from vaccination to new infection not addressed. Again, this has been included as above for comment 1.

Study 3

RE: comment 1; classification of unvaccinated. The authors have explained the difference between the 7 and 14 day classification of individuals as unvaccinated in study 1, comment 1. There is no requirement for the data to be included within the main text, as long as it is provided within the supplementary (Table S12).

REVIEWERS' COMMENTS

Reviewer #1 (Remarks to the Author):

I agree with the changes made, except for the presentation of the models in the appendix:

The time-dependent variables $T_i(t)$ and $T_{Vi}(t)$ and $T_{V_C_i}(t)$ are correctly presented in models 1.5, 1.6 and 1.7. But in models 1.8, 2.1 and 2.2 the (t) is still lacking. Correct presentation will help the readers understand the analyses done. Line 174 still includes an '= 1'.

We thank the Reviewer for their thorough and thoughtful comments. We have made the corrections for models 1.8, 2.2, and 2.2 in the Appendix/Supplementary Information. The = 1 corresponding to the description of variables in Model 2.2 were an error and have been removed.

Reviewer #3 (Remarks to the Author):

This is a review of the manuscript "Effectiveness and Protection Duration of Covid-19 Vaccines and Previous Infection Against Any SARS-CoV-2 Infection in Young Adults" (NCOMMS-21-50698A). Overall, it is a comprehensive paper discussing the vaccine effectiveness (and the duration of this protection) in university students during the delta wave, split by vaccine manufacturer as well as those with/without prior infection and how waning impacts protection against disease.

However, this is a review in place of reviewer #2, to ensure their comments have been covered by the authors in their revised article. In general, all comments by reviewer #2 have been covered or rectified by the authors. Further details are below:

Study 1

RE: comment 1; 14 or 7-day window classification as unvaccinated. This is the biggest concern reviewer 2 had about the article, detailing that a difference between 7 and 14 days may be observed when classifying individuals moving from the unvaccinated > vaccinated group, with a time variable. However, I will agree with the authors that the literature is confounded when classifying individuals as unvaccinated (some use the day of vaccine, whilst others use up to 21 days). Nevertheless, the work the authors have detailed shows that there is a minimal difference when classifying individuals as unvaccinated for 7 or 14 days. The authors have added further clarification to their method and results sections covering this. I do not believe a table needs adding to the main text, the addition of a new section to their results ("Classification of Vaccination Status") summarises this data well and provides the reader with an explanation of the time variable used and how variation by days does not alter the results.

RE: comment 2; time passed since 2nd dose. The addition of a further paragraph within the discussion satisfies reviewer #2 comments, and expands on the limitations (e.g. not individual-level protection).

Study 2

RE: comment 1; adjusting for time since dose. Again, the authors have shown that their analysis has been sufficient to answer this question, detailing minimal differences between their original estimated protection or sensitivity analysis. The authors have included this data in model 1.8 and table S13.

RE: comment 2; time passed from vaccination to new infection not addressed. Again, this has been included as above for comment 1.

Study 3

RE: comment 1; classification of unvaccinated. The authors have explained the difference between the 7 and 14 day classification of individuals as unvaccinated in study 1, comment 1. There is no requirement for the data to be included within the main text, as long as it is provided within the supplementary (Table S12).

We thank the Reviewer for their assessment of this manuscript. Based on these comments, it appears that we have sufficiently addressed Reviewer #2's original concerns and further action is not necessary.